# *Ascophyllum nodosum* (Linnaeus) Le Jolis from Arctic: Its Biochemical Composition, Antiradical Potential, and Human Health Risk

**DOI:** 10.3390/md22010048

**Published:** 2024-01-19

**Authors:** Ekaterina D. Obluchinskaya, Olga N. Pozharitskaya, Elena V. Gorshenina, Anna V. Daurtseva, Elena V. Flisyuk, Yuliya E. Generalova, Inna I. Terninko, Alexander N. Shikov

**Affiliations:** 1Murmansk Marine Biological Institute of the Russian Academy of Sciences (MMBI RAS), 17 Vladimirskaya Str., 183038 Murmansk, Russia; olgapozhar@mail.ru (O.N.P.); gev1811@yandex.ru (E.V.G.); tav.mmbi@yandex.ru (A.V.D.); spb.pharmacy@gmail.com (A.N.S.); 2Department of Technology of Pharmaceutical Formulations, St. Petersburg State Chemical Pharmaceutical University, 14 Prof. Popov Str., 197376 Saint-Petersburg, Russia; elena.flisyuk@pharminnotech.com; 3Core Shared Research Facilities “Analytical Center”, St. Petersburg State Chemical Pharmaceutical University, 14 Prof. Popov Str., 197376 Saint-Petersburg, Russia; generalova.yuliya@pharminnotech.com (Y.E.G.);

**Keywords:** Arctic, *Ascophyllum nodosum*, polyphenols, toxic metals, antioxidants, fucoidan, mannitol, alginic acid

## Abstract

*Ascophyllum nodosum* is a brown seaweed common in Arctic tidal waters. We have collected *A. nodosum* samples from the Barents Sea (BS), Irminger Sea (IS), and Norwegian Sea (NS) in different reproductive stages and have evaluated their biochemical composition, radical scavenging potential, and health risks. The total content of dominating carbohydrates (fucoidan, mannitol, alginate, and laminaran) ranged from 347 mg/g DW in NS to 528 mg/g DW in BS. The proportion of two main structural monosaccharides of fucoidan (fucose and xylose) differed significantly between the seas and reproductive phase, reaching a maximum at the fertile phase in the BS sample. Polyphenols and flavonoids totals were highest in NS *A. nodosum* samples and increased on average in the following order: BS < IS < NS. A positive correlation of free radical scavenging activity for seaweed extracts with polyphenols content was observed. The concentration of elements in *A. nodosum* from the Arctic seas region was in the following order: Ca > Mg > Sr > Fe > Al > Zn > As total > Rb > Mn > Ba > Cu > Co. Seaweeds from BS had the lowest metal pollution index (MPI) of 38.4. *A. nodosum* from IS had the highest MPI of 83. According to the calculated target hazard quotient (THQ) and hazard index (HI) values, Arctic *A. nodosum* samples pose no carcinogenic risk to adult and child health and are safe for regular consumption. Our results suggest that the Arctic *A. nodosum* has a remarkable potential for food and pharmaceutical industries as an underestimated source of polysaccharides, polyphenols, and flavonoids.

## 1. Introduction

Brown algae from the Arctic region are a promising source of extraordinary compounds that can be used to produce various products with beneficial properties. Brown algae of the Fucaceae family, such as *Fucus*, *Ascophyllum*, and *Pelvetia*, dominate in intertidal zones of numerous regions of the Northern Hemisphere [1,2]. *Ascophyllum nodosum*, commonly known as rockweed, is a brown algae belonging to the family Fucaceae of the order Fucales (class Phaeophyceae, phylum Ochrophyta). *A. nodosum* is a common fucoid algae found in sheltered, intertidal, and rocky shores Arctic rocky shores. The genus is monotypic and basal in Fucaceae and is found from Arctic Canada, Greenland, Iceland, and northern Norway to the southern reaches of Portugal and Long Island (USA) [3]. Natural observations on the Barents Sea coast showed that Ascophyllum is mainly found in the littoral baths (salinization zones) [4]. Ascophyllum is used in the production of alginate, fertilizers, animal nutrition, and cosmetics. There are approximately 50 commercial companies that manufacture *A. nodosum* extracts for use in agriculture and horticulture [5]. *A. nodosum* is a microalgae that is rich in diverse bioactive phenolic compounds, such as phlorotannins, and unique polysaccharides, including alginic acid (15–30%), fucoidans (4–10%), mannitol (5–10%), and laminaran (0–10%) [6,7,8,9]. It is also a nutritionally rich source of protein (5–10%), lipids (2–7%), carbohydrates (40–70%), ash (15–25%), and other compounds [9,10,11]. *A. nodosum*, similarly to other brown seaweeds, contains different carotenoids including fucoxanthin, fucoxanthinol, zeaxanthin, violaxanthin, α- and β-carotene, and others [12]; chlorophyll [13]; anthocyanin [13]; simple phenolics like catechin and epigallocatechin [14,15]; hydroxybenzoic acid; coumaric acid; cinnamic acid; rosmarinic acid; and caffeic acid [15]; amino acids [16]; etc. However, some of these compounds exhibit significant seasonal variation [17,18].

Seaweed-derived fucoidans have been extensively studied for their anti-inflammatory [19,20], anticoagulant, antiangiogenic [19], and antioxidant properties [21]. In ex vivo experiments, fucoidans have been shown to protect elastic fibers from proteolytic enzymes [22]. These results suggest a possible use of fucoidans in the therapy of the joints inflammation.

Fucoidans from *A. nodosum* have been shown to prevent degradation of elastic fibers by minimizing human leukocyte elastase [23], stimulation of dermal fibroblast proliferation [24], acceleration of collagen processing by cultured fibroblasts [24], and antioxidant properties [8,25,26,27].

Antioxidant, anti-inflammatory, and anti-ageing properties were reported for phlorotannins-rich extract of *A. nodosum* [28]. Several scientists have explored the potential of brown algae in cosmetics. Inhibition of tyrosinase [25], anti-elastase effects [29], and protection of skin cells against UVB-induced mutations and chronic inflammation [30] has been associated with the presence of phlorotannins in extracts of *A. nodosum*. Phlorotannins have been extensively studied for their anti-diabetic activities [31]. These compounds inhibit α-amylase and α-glucosidase enzymes, etc. [26].

However, Arctic samples of *A. nodosum* collected at different reproductive phases were not yet systematically assessed. In this study, we aimed to evaluate a biochemical composition, radical scavenging potential, and health risks of *A. nodosum* collected for different reproductive phases and different Arctic locations.

## 2. Results and Discussion

### 2.1. Carbohydrates Composition

The content of fucoidan in the tested samples of *A. nodosum* varied from 119.0 ± 8.9 mg/g dry weight (DW) from the Cape Sydspissen of the Norwegian Sea (NS) to 204.8 ± 8.4 mg/g DW in the Zavalishina Bay of the Barents Sea (BS) (St. 5) (Figure 1). The mean content of fucoidan in *A. nodosum* samples from the Irminger Sea (IS), NS, and BS was 134.0 ± 21.0, 128.5 ± 14.6, and 159.6 ± 35.8 mg/g DW, respectively). There difference in fucoidan content in samples from Arctic seas was not statistically significant. While the levels of fucoidan in *A. nodosum* collected at different locations in BS (St. 4 and St. 5) differed significantly. Although water temperature and salinity did not affect the accumulation of fucoidan, its level in the sterile reproductive phase was slightly lower (Figure 2).

The fucose and xylose contents n *A. nodosum* from Arctic seas collected in different productive phases analyzed HPLC-RID are shown in Table 1.

The content of fucose ranged from 59.5 mg/g DW in the samples from IS (St. 1) to 102.4 mg/g DW in seaweeds from BS (St. 5). Fucose is the main monosaccharide of fucoidan from *A. nodosum*. Its content in fucoidan varies from 40% [32] to 58.5% [33]. One of the minor monosaccharides is xylose, which is always part of fucoidan from *A. nodosum* [34]. In a study [35], polysaccharides were extracted from different parts of *Marginariella boryana* separately. The vegetative structures (blades and vesicles) and reproductive structures (receptacles) were found to have different proportions of minor monosaccharides such as xylose, mannose, and uronic acid. The vegetative organs had significantly higher proportions of these minor monosaccharides. The ratio between major (fucose) and minor (xylose) monosaccharides could be useful in understanding the accumulation of fucoidan during the reproductive phase [36,37].

Our study revealed that the samples from the BS contained the highest amount of fucose (Table 1). We also found that fucose accumulates more actively during the fertility phase, while xylose accumulates more during the sterile phase. The accumulation of fucose and xylose by *A. nodosum* was negatively correlated with water salinity (Pearson’s correlation coefficients *r* = −0.469, *p* < 0.05 and *r* = −0.487, *p* < 0.05, respectively). We did not observe any relationship between water temperature and fucose level. However, it was found that xylose accumulates more actively in cold water. There was a positive correlation (Pearson’s correlation coefficients *r* = 0.503, *p* < 0.05) between fucose content and reproductive phase. Relatively uniform fucose-rich fucoidan was produced by *A. nodosum* from the Barents Sea in both sterile and fertile phases. The amount of fucose present varied greatly depending on the reproductive cycle.

The amount of biologically active matabolites in algae vary depending on the season and reproductive phase, respectively. Brown seaweeds from the Far East and Arctic regions accumulate the highest amount of polysaccharides during sporulation. The composition of algae monosaccharides changes in accordance with the reproductive phase [38,39]. It was shown that during sporulation, the proportion of the main monosaccharide fucose increased significantly, and other minor monosaccharide mannose decrease [39]. Fucose percentage in the polysaccharide was also increased in the reproductive phase in Japanese kelp [40], *F. vesiculosus* [36], and *F. spiralis* [37].

The mannitol content in *A. nodosum* specimens was 58.0–97.9 mg/g DW (Figure 1). Samples from the IS and NS were more rich in mannitol comparing with samples from the BS. There was a significant effect of the reproductive phase on mannitol levels in samples from NS, but not from IS and BS (Figure 2). Mannitol content in algae was positively correlated with water salinity (*r* = 0.659, *p* < 0.05).

Mannitol is the reserve metabolite of brown seaweeds. Its accumulation depends on seaweed species, seasons, and growing location. Due to its wide application in industry, mannitol is in demand. Isolating it from algae is cheaper than synthesizing it chemically [41]. The genus Laminaria is the richest source of mannitol, with up to 20–30% of DW of biomass [42]. The mannitol content in *A. nodosum* from the Dalnezelenetskaya Bay of the Barents Sea could be up to 6.8% DW [43] and to 10.9% DW from the Rebalda of the White Sea [44]. Mannitol content in *F. vesiculosus* was positively correlated with the water salinity [45]. During summer and autumn, brown seaweed is known to accumulate carbohydrates and mannitol, with levels decreasing in winter [46,47]. A study conducted on fresh seaweed samples from Thorverk (Iceland) found that the mannitol content was significantly lower in June (7.1 ± 0.3% DW) compared to July and October [48].

The amount of alginic acid present in the tested samples of *A. nodosum* varied from 70.1 ± 4.8 mg/g DW in the Ringvassøya Island of the NS (St. 2) to 199.3 ± 3.7 mg/g DW in the Zavalishina Bay of the BS (St. 5) (as shown in Figure 1). The alginic acid content also depended on the location of algae collection (Pearson’s correlation coefficients *r* = 0.370, *p* < 0.05). For Teriberskaya Bay in the Barents Sea, the values varied from 129.9 ± 1.0 (St. 4) to 199.3 ± 3.7 mg/g DW (St. 5). A correlation was found between the reproductive phase (Pearson’s correlation coefficients *r* = 0.456, *p* < 0.05) (Figure 2) and salinity (Pearson’s correlation coefficients *r* = −0.528, *p* < 0.05) with the content of alginic acid. However, no correlation was found between the content of alginic acid and water temperature. The level of alginic acid from *A. nodosum* collected in autumn in Dalnezelenetskaya Bay was slightly lower and reached 128.2 ± 11.6 mg/g DW [39]. According to reference [49], Fucus collected from Avacha Bay in Kamchatka contained 173 mg/g DW of alginic acid. High amount s of alginic acid (28% DW) were found in brown algae in Irish waters [50] and 23% DW in *A. nodosum* from coast of Shetland [11].

Laminaran content in the tested samples ranged from 47.7 to 102.2 mg/g DW (Figure 1). In this study, it was observed that laminaran accumulated more actively during the sterility phase compared to the fertility phase (Figure 2). It was found that there is a relationship between the amount of laminaran present and the reproductive phase (Pearson’s correlation coefficient of *r* = 0.768 and *p* < 0.05). Positive correlation was observed between laminaran accumulation and salinity (Pearson’s correlation coefficient of *r* = 0.452 and *p* < 0.05). On the other hand, a negative correlation was found between laminaran content and temperature (Pearson’s correlation coefficient of *r* = −0.556 and *p* < 0.05). In Irish waters, brown seaweeds can contain up to 4.5% of laminaran in their dry weight [45]. The level of laminaran from *A. nodosum* collected in autumn in Dalnezelenetskaya Bay of the BS was 5.1 ± 0.3% DW [39]. Laminaran is the principal reserve polysaccharide and is composed of β-glucan [46]. Algae produce more laminaran during winter to survive when photosynthesis is limited [47,48]. The amount of laminaran present in algae depends on various factors such as the species of algae, growth phases, and the method of extraction used. For instance, the Kamchatka brown algae contain 10–11% laminaran [44], while the brown algae found in the BS has a laminaran content ranging from 0.6 to 11% of DW. Arctic brown algae tend to accumulate mannitol and laminaran during short summer, with their maximum contents by the end of season. This is followed by a significant decline in long and cold winter season to support survival [49]. Similar results were obtained for brown algae found in the seas of the Arctic (Figure 2). These algae appear to experience a shortage of photosynthetic carbon during their growth phase, which they may compensate for by reusing their stored carbohydrates [51].

The laminaran content was from 47.7 to 102.2 mg/g DW in the studied *A. nodosum* samples (Figure 1). It was observed that laminaran accumulated more actively in the infertile phase compared to the fertile phase (Figure 2). It was found that there is a relationship between the amount of laminaran present and the reproductive phase (Pearson’s correlation coefficient of *r* = 0.768 and *p* < 0.05). A positive correlation was observed between laminaran accumulation and salinity (Pearson’s correlation coefficient of *r* = 0.452 and *p* < 0.05). A water temperature negatively correlated with laminaran content with the Pearson’s correlation coefficient *r* = −0.556 and *p* < 0.05. In Irish waters, brown seaweeds can contain up to 4.5% of laminaran in their dry weight [50]. The level of laminaran from *A. nodosum* collected in autumn in Dalnezelenetskaya Bay of the BS was 5.1 ± 0.3% DW [43]. Laminaran is the principal reserve polysaccharide [51]. Algae produce more laminaran during winter to survive when photosynthesis is limited [52]. The amount of laminaran present in algae depends on various factors such as the species of algae, growth phases, and the method of extraction used. For instance, the Kamchatka brown algae contain 10–11% laminaran [49], while the brown algae found in the BS has a laminaran content ranging from 0.6 to 11% of DW. Perennial brown algae in the Northern Hemisphere tend to accumulate mannitol and laminaran during the summer months, resulting in their maximum contents by late summer and early autumn, with levels decreasing significantly in winter to support growth [53,54]. Similar results were obtained for brown algae found in the seas of the Arctic (Figure 2). These algae appear to experience a shortage of photosynthetic carbon during their growth phase, which they may compensate for by reusing their stored carbohydrates [55].

### 2.2. Polyphenols and Flavonoids Content

The total polyphenols content (TPhC) in *A. nodosum* varied widely across different geographic locations, ranging from 31.4 to 95.4 mg of phloroglucinol equivalent per 1 g of DW weight (Figure 3). The total flavonoid content (TFC) was approximately 5.8 times lower when compared with TPhC.

The study found that the highest concentration of phenolic compounds (as the sum of polyphenols and flavonoids) was present in *A. nodosum* collected in Norwegian sea, with the following ranking order: BS < IS < NS. Upon analyzing the sterile–fertile patterns from IS (St. 1), NS (St. 3), and BS (St. 5), it was observed that seaweeds in the fertile phase had lower levels of polyphenols and flavonoids compared to the sterile phase. The ratio of polyphenols to flavonoids was also significantly lower in the fertile phase. The multifactorial ANOVA test revealed that the reproductive phase and geographical location had a significant impact on TPhC (*p* < 0.05) (Figure 3).

To be secondary metabolites, phenolic compounds are not directly involved in photosynthesis, cell division, or production of alga. Phenolics are stress compounds that protect against biotic and abiotic factors such as grazing, bacteria, UV radiation, and metal contamination [56]. Seasonal variability in chemical composition is common for perennial algae such as *A. nodosum* and is associated with physiological changes within vegetative thalli, as shown by P. Åberg and H. Pavia (1997), seasonal variability in chemical composition is common for perennial algae such as *A. nodosum* and associated with physiological changes within vegetative thalli. In addition to the seasonal occurrence of reproductive structures (the time of which varies according to geographic location and shore level [57]. In this study, we investigated the variation in phlorotannin content in *A. nodosum* thalli from the IS, NS, and BS depending on the reproductive stage of the algae. Due to the potential use of *A. nodosum* in food and cosmetics or medicine, it is necessary to determine the most favorable time to collect algal material. Significant seasonal changes in the polyphenol content of *A. nodosum* were described by S. Parys et al. (2009), E. Apostolidis et al. (2011), and M.R. Tabassum et al. (2016). The polyphenol content in the seaweed is dependent on the location, harvesting time, light intensity, temperature, salinity, and ambient nutrients [18]. Polyphenols peaked in June (4.9% DW) and were lowest in April (0.2% DW) [58]. Apostolidis et al. (2011) reported a similar seasonal variation in phenol content, with June and July having the highest and May the lowest [59]. The reason for the variation in polyphenol content may be location, light intensity, temperature, salinity, and ambient nutrients [18]. The reproductive stage of the seaweed also significantly affects the variations in phenol content. Comparatively, lower polyphenol concentrations during the fertile period (April–June) were recorded than the time of shedding fruit bodies at the end of June [60]. Throughout the year, *A. nodosum* goes through different phases of growth and development. It is fertile from April to June, and at the end of June, fruit bodies are shed. In September, the development of conceptacles at the end of the short shoots starts [61]. In previous studies, it was found that the concentration of polyphenols and flavonoids in brown algae species F. vesiculosus and F. spiralis from the Arctic region is lower during the fertility phase. However, the maximum concentration is achieved during the period when the fruiting bodies are shed [36,37]. Changes in polyphenol content correlate with *A. nodosum*’s reproductive stage.

### 2.3. DPPH Radical Scavenging Activity

The concentration of seaweed extract required to scavenge 50% of the DPPH radicals was expressed as IC_50_ for the radical scavenging activity of seaweed extracts. The IC_50_ values ranged from 0.35 mg/mL to 1.0 mg/mL (Figure 4). The lowest values were found in the algae samples from IS (St. 1) and from NS (St. 3) collected during the sterile phase, while the highest value was observed in the sample (St. 5) from BS during the fertile phase. In our study, the IC_50_ decreased during the sterile phase compared to the fertile phase. This decrease could be associated with an increase in TPhC and TFC during the sterile phase (Figure 3). The contents of TPhC and TFC significantly affects IC_50_ (Pearson’s correlation coefficient *r* = −0.737, *p* < 0.05 and *r* = −0.639, *p* < 0.05).

In this study, IC_50_ positively correlated with fucoidan content (*r* = 0.810, *p* < 0.05). This fact suggests that the anti-radical activity of *A. nodosum* extracts is negatively affected by the fucoidan content. Although previous studies have reported radical scavenging activity for fucoidan, they used crude fucoidan [62]. It is possible that other compounds commonly found as admixtures in crude fucoidans, such as minor phenolic compounds, ascorbic acid, fucoxanthin, proteins, etc., may influence anti-radical activity. In a previous study, polysaccharide fractions rich in sulfate and containing small amounts of polyphenols was found to have very weak antioxidant activity. According to T. Imbs et al. (2015), polyphenols that were co-extracted with the sulfated polysaccharides, contributed to anti-radical activity of sulfated polysaccharides from Fucus algae from the Okhotsk Sea [63]. The amount of phenolic compounds (TPhC and TFC) present in the samples of *A. nodosum* was found to be the main factor determining their radical-scavenging activity. Pearson’s correlation coefficients showed a significant negative correlation between the TPhC and TFC with the radical scavenging activity of the samples (*r* = −0.737 and *r* = −0.639, respectively, *p* < 0.05).

Radical scavenging effects of *A. nodosum* polyphenols have been studied in vitro in several works. It has been shown that the accumulation of polyphenols correlates with activity against DPPH and other radicals [29,64,65]. Our results are in line with previous literature data. Several scientific publications have evidence about direct correlation between DPPH scavenging and TPhC in algal extracts [20,63,66]. Flavonoids also contribute to the DPPH scavenging activity. During high and low tides, littoral algae are affected by extreme conditions such as dryness, air, and UV radiation exposure. In response to oxidative stress, seaweeds produce antioxidants, including polyphenols and flavonoids. Scientific publications confirm that sea algae of all classes produce compounds with antioxidant activity [67]. We found that water salinity and region sampling affect the radical scavenging activity of *A. nodosum* (F = 8.59, *p* = 0.043 and F = 10.67, *p* = 0.043).

### 2.4. Contents of Elements

The concentrations of elements in each sample of *A. nodosum*, with their minimum and maximum ranges and the LOQ, are shown in Table 2. The concentrations of elements differed based on the location and reproductive phase of the seaweed. Algae from IS accumulated elements to a greater extent than algae from NS and BS. Moreover, the concentration of elements depended on salinity (Pearson’s correlation coefficients *r* = 0.640, *p* < 0.05), reproductive phase (Pearson’s correlation coefficients *r* = 0.623, *p* < 0.05), sampling sites (Pearson’s correlation coefficients *r* = 0.705, *p* < 0.05), and temperature (Pearson’s correlation coefficients *r* = −0.301, *p* < 0.05). Ca concentration averaged 15,595 mg/kg DW and maximal (23173 mg/kg DW) in samples from NS (St. 3) and 26,916 mg/kg DW in samples from IS (St. 1) during the sterile reproductive period. Fluctuations in Ca concentration in samples from BS were insignificant; on average, it was 12,092 mg/kg DW in the fertile phase and increased to 13,202 mg/kg DW in the fertile phase. The Mg concentration was lower, averaging 9970 mg/kg DW. All samples had high Sr concentrations. Several toxic elements (Pb, Cd, Cr, or Ni) were not detected in most samples of *A. nodosum* from various seas of the Arctic region or were below the LOQ. The accumulation of fucoidan in *A. nodosum* samples was found to be inversely correlated with the concentration of the analyzed elements. Alginate content in algae samples did not correlate with elemental accumulation, while laminaran was positively correlated with elemental concentration. The concentration of elements in algae of the seas of the Arctic region can be arranged in descending order of average values: Ca > Mg > Sr > Fe > Al > Zn > As total > Rb > Mn > Ba > Cu > Co. Similar results for contents of elements in other Arctic brown seaweeds were published in the literature [36,37].

Macrophytes contain almost all the elements common in seawater. Their ratios in algae vary significantly among different species. Algae have a selective cumulative ability, as a result of which a diverse complex of microelements accumulates in their thalli, and the concentration of some of them in tissues, like calcium, can be tens of times higher; for bromine and chromium, it is hundreds of times; and for elements like iodine, zinc, and barium, it is even thousands of times higher than in seawater [68]. The mechanisms of metal accumulation by algae have not complete studied. However, these mechanisms are associated with a high content of polysaccharides, which involved in ion exchange processes that actively occur in algal polysaccharides [68]. L. Andrade et al. (2004), when studying the ultrastructure of acidic polysaccharides from the cell walls of brown algae *Padina gymnospora*, showed that Zn is associated with Ca and S (in addition to C and O), has been indicated that alginic acid and sulfated fucan are molecules responsible for binding heavy metal [69].

P. Mariani et al. [70] studied the elemental composition of polysaccharides isolated from *Fucus virsoides* and suggested that Ca^2+^ and Mg^2+^ form strong complexes with alginic acid in algal cell walls, stabilizing the wall structure. In contrast, seawater cations were mainly associated with sulfated polysaccharides, which regulate passive ions found in algal cells. L. Andrade et al. showed that acidic polysaccharides (mainly sulfated fucans) play an important role in the nucleation of Cd and Zn [71]. Thus, polyanionic polysaccharides perform a binding function with metals for the nucleation and precipitation of heavy metals in the cell walls of brown algae.

A. Jensen et al. [72] have suggested that metal concentrations in algal tissue could be used as a criterion for pollution in certain marine regions. Several studies were conducted on *A. nodosum* from Lofoten (Norway), and basic concentrations in dried algal tissue were considered to be 75 mg Zn/kg DW, 5.5 mg Cu/kg DW, <3 mg Pb/kg DW, and <0.7 mg Cd/kg DW. However, *A. nodosum* concentrations in Trolla (Bænnebukta, Trondheimsfjorden) were very high for Zn (375–700 mg/kg) and Cu (18–60 mg/kg) [72]. This result indicates that the concentrations of Cu, Pb, Cd, and Zn in *A. nodosum* in the present study were low compared to previous studies on the same species by both from NS and from IS and BS [72].

Arsenic (As) in forms of arsenate anion (HAsO_4_^2−^) instead of a phosphate anion (HPO_4_^2−^) could be accumulated by seaweeds from the water. Even though it gets metabolized into arsenosugars and arsenolipids, which are organoarsenic compounds, it can still be present in seaweed biomass. It poses a potential health risk for people who consume seaweed products. Previous studies have revealed that the concentration of inorganic arsenic (iAs) in *A. nodosum* is less than 1% of the total arsenic (As) content [73]. Here, total As concentration was determined in intertidal fucoid *A. nodosum* using inductively coupled plasma optic emission spectrometry (ICP-OES). The concentration of total As in the samples varied slightly and had an average of 32.2 ± 7.9 mg/kg DW (Table 2). According to the literature data, the total As in Phaeophyta spp. is varied from 1.9 to 245.2 mg/kg DW. While in Rhodophyta, Phaeophyta, and Chlorophyta algae, arsenicum concentration is about 30, 100, and 20 mg/kg DW, respectively [74].

Certain compounds found in algae have metal-binding sites that are beneficial to humans. The elements can be freely absorbed from the surrounding seawater by the polysaccharides of algae cells [75]. Alginates from brown algae have a higher affinity for binding heavy metals compared to carrageenans and agar from red algae, according to binding affinity estimates [76]. The correlation between metals content and levels of alginic acid and fucoidan in tested *A. nodosum* samples was weak (Pearson’s correlation coefficients *r* = −0.166, *p* < 0.05, and *r* = −0.424, *p* < 0.05 for alginic acid and fucoidan, respectively). However, we noted a positive correlation between the levels of laminaran and mannitol and total concentration of metals.

Brown algae phenols can chelate heavy metals with widely varying affinities [77]. Such properties may be important because the phenolic contents of seaweeds increase with salinity. The element concentrations found in *A. nodosum* from the West Greenland fjord system [78] are generally lower than those from northern Europe but are similar to those in Iceland. The concentrations of Zn, Pb, and Fe in algae are considerably lower in Greenland.

The reproductive phases and seasons affect the accumulation of some metals in *A. nodosum*. Table 2 shows that the levels of As, Ba, Ca, Rb, Sr, and Zn are lower in the fertile phase. It could be due to the growth of seaweeds. Previous studies have shown that trace elements tend to accumulate in seaweeds during dormant periods in winter and decrease during the summer months when they grow and reproduce [79,80].

The seasonal variation of certain metals in algae is closely associated with the bioavailability of metals from seawater. The salinity of sea water increases in the summer season (fertility phase of seaweeds) due to rare rains and increased evaporation. This leads to an increase in Cd, Zn, and Cu binding to chloride ions [81]. In samples of *A. nodosum* from the IS, the total accumulation of metals was 1.3–1.4 times higher than in algae from the coasts of the NS and BM (Table 2). It indicates that seawater from the NS and BS coasts had lower bioavailability of metals.

Ca and Mg intake correlates with cardiovascular health. It has been suggested that appropriate intake of magnesium may lower blood pressure because it acts as a Ca antagonist on smooth muscle tone, thereby causing vasorelaxation [82]. Data presented in Table 2 indicate that *A. nodosum* tends to accumulate more Ca than Mg. A similar trend is typical for brown algae [83]. It should be emphasized that the Ca/Mg ratio is also significant for calcium absorption, since insufficient magnesium intake can lead to excessive accumulation of calcium in soft tissues, leading to the formation of kidney stones and arthritis [84]. High Ca/Mg intake ratios have been associated with increased risk of cardiovascular disease, colorectal and prostate cancer, and overall cancer mortality [85,86]. Evidence suggests that a Ca/Mg ratio range of 1.70 to 2.60 may reduce the risk of disease [86]. However, these data are still limited, as only a few studies have analyzed dietary Ca/Mg ratios <1.70 as well as >2.60 [86]. Moreover, these benefits may depend on specific health indicators and gender [86]. Macroalgae from the present study showed Ca/Mg ratios ranging from 1.08 in algae collected from the Dalnezelenetskaya Bay of the Barents Sea to 2.67 in algae from the Irminger Sea (Table 2). For algae from the Irminger Sea and the Norwegian Sea, a correlation the reproductive phase on the Ca/Mg ratio has been established. Only one sample of *A. nodosum* collected in the sterile phase had a Ca/Mg ratio in the optimal range of 1.70–2.60. Therefore, this sample could be recommended for food supplementation to help balance Ca/Mg ratio.

### 2.5. Metal Pollution Index

The metal pollution index (MPI) for *A. nodosum* samples collected at different regions of the Arctic are presented in Figure 5.

The generalized mean MPI for *A. nodosum* samples collected in different Arctic seas was 56 and was increased in the following order: BS < NS < IS (Figure 5). The MPI was depended on the seaweed location (Pearson correlation *r* = 0.69, *p* < 0.05), whereas the effect of reproductive phase on MPI was weak (Pearson coefficient *r* = 0.27, *p* < 0.05). The MPI value in the fertile stage was significantly higher for samples from the NS and BS, while the MPI for samples from the IS was practically independent of the reproductive phase. The water salinity and temperature positively correlated with MPI (*r* = 0.55, *p* < 0.05 and *r* = 0.69, *p* < 0.05, respectively).

The quality of coastal marine water based on metals can be classified by the following recommendations [87,88]. It was suggested earlier that *A. nodosum* could be used for monitoring seawater metal contamination [89]. Based on the above-mentioned recommendations and our analysis of metal concentrations in *A. nodosum*, we can conclude that the seawater around sampling places in the Barents, Irminger, and Norwegian seas during August–September 2019 can be classified as “Class I—Unpolluted”.

### 2.6. Human Health Risk

The mean and maximum concentration, the daily dose, and a comparison with the risk estimations for a 70 kg man [90,91,92,93] and with Nutritional Requirements [90,94,95] are presented in Table 3 for every element detected in *A. nodosum*.

The consumption of seaweed has been rising in Western countries in recent years. Seaweed is a nutritious food source due to its high protein, fatty acid, vitamin, and mineral content. It is becoming increasingly popular to include seaweed in daily diets [96]. *A. nodosum* accumulates nutrients and minerals from the seawater, and this makes the alga a valuable resource for industry [3,97]. It is harvested in Ireland, Scotland, Norway, and Russia for the manufacture of seaweed meal [3] and consumed in Iceland and Greenland, and it is also used to make herbal tea [98,99]. Due to the content of vitamins, trace elements, lipids, carbohydrates, proteins, etc., *A. nodosum* can be recommended as a dietary supplement. In 2019, the global production of seaweeds amounted to around 35 million tons [100]. Regulations for limiting toxic elements in algae are different for each country. France has approved upper limits for Pb, Cd, Sn, Hg, As, and I in seaweed for human consumption [101]. In Russia, there are established limits for the presence of Pb, As, Cd, and Hg in algae [95]. The potential human toxicity of Arctic *A. nodosum* was assessed in the current study based on FAO/WHO [80,81,82], and European [90] documents.

For the essential elements (Table 3), compared to the respective ULs reported by EFSA [102], the daily intake of 3.3–12.5 g of *A. nodosum* corresponds to about 8–13.6% of the daily tolerated amount of Ca; about 2% of the tolerable daily dose of Cu, about 7.5% of the daily tolerable dose of Zn, and approximately 1–3% of the daily tolerable dose of Al [103]. Daily intake of 3.3–12.5 g of *A. nodosum* corresponds to a total daily intake of 267–386% of the tolerable daily dose of inorganic As or 1–1.5% of the tolerable daily dose of total As. In this study, As was measured as total As in all *A. nodosum* samples. Inorganic As causes several dangerous diseases [104,105] and is classified as a carcinogen (Group 1) [106]. It is noteworthy that arsenic in algae exists predominantly in organic complexes (especially with sugars), which are considered significantly less toxic than inorganic arsenic [107].

Based on the potential daily consumption of seaweed and the concentrations of elements found in *A. nodosum* from seas of Arctic, the health risk for children and adults was calculated. Algae are known to accumulate more elements than aquatic plants/organisms. Therefore, regular consumption of algae may pose a particular risk. Figure 6 shows the estimated non-carcinogenic and carcinogenic risks associated with metal intake that could result from consumption of *A. nodosum* seaweed.

Generally, health risks associated with an element cannot be predicted if the target hazard quotient (THQ) value is less than 1 [108,109]. In the present study, all elements in *A. nodosum* samples had THQs less than 1. This indicates that there is no potential health risk to humans; HI values were also less than 1 (Figure 6a). The average HI value of all algal samples was 0.50 for adults and 0.22 for children. As, Cd, Ni, Cr-VI, and Pb metals are all known potential carcinogens [110]. Since the concentrations of Cr, Pb, Cd, and Ni metals were below the LOQ in all algal samples (Table 3), the carcinogenic risk was assessed in terms of As intake. The lifetime cancer risk (LTCR) from exposure due to total lifetime As intake from seaweeds is summarized in Figure 6b. For lifetime seaweed consumption and associated dietary As exposure, total cancer risk was calculated as an average of 0.182 (range 0.163–0.218) for children and 0.400 (range 0.359–0.478) for adults according to the US EPA [111], with LTCR values below 10^−6^ considered negligible, while risk between 10^−6^ and 10^−4^ is considered acceptable. According to this study, the estimated As exposure does not lead to a cancer risk from the lifetime consumption of the studied seaweeds.

It is important to note that *A. nodosum* collected in the Barents, Irminger, and Norwegian seas does not accumulate dangerous concentrations of toxic elements. These algae can be recommended to enrich the daily diet.

## 3. Materials and Methods

### 3.1. Samples Collection

Samples of *A. nodosum* were collected in the coastal zones (low tide at 0.6–1.0 m depth) of the Irminger Sea (IS), Norwegian Sea (NS), and Barents Sea (BS) in 2019. The harvesting procedure is described in detail in previous publications [112]. *A. nodosum* samples were collected in Iceland, Norway, and Russian Aquatopia’s. Details are described in Table 4.

### 3.2. Chemicals and Reagents

DPPH (2,2-diphenyl-1-picrylhydrazyl), Folin–Ciocalteu reagent, fucose, glucose, phloroglucinol, quercetin, and xylose were all purchased from Sigma-Aldrich (St. Louis, MO, USA). All other analytical-grade chemicals and solvents were purchased from local chemical suppliers. Multi-Element Calibration Standard 3 for elements analysis was from PerkinElmer, Shelton, CT, USA.

### 3.3. Carbohydrates Composition

For the determination of fucoidan content, seaweed samples were processed according to the procedure [49]. Fucoidan content was determined by the cysteine-sulfuric acid method [113], and *L*-fucose was reference.

Fucoidan for carbohydrate analysis was hydrolyzed with 2 M trifluoroacetic acid (0.5 mL) at 121 °C for 2 h. The samples were then cooled in an ice bath and centrifuged at 5000 rpm for 5 min and the liquid fraction was adjusted to pH 7 with 2 M NaOH [114]. Monosaccharides were determined using high-performance liquid chromatography (HPLC Model LC 20 AT Prominence, Shimadzu, Kyoto, Japan) with a refractive index detector (RID-10A, Shimadzu, Kyoto, Japan) [115].

Mannitol level in *A. nodosum* samples was measured at 597 nm (Shimadzu UV 1800, Shimadzu, Kyoto, Japan, Japan) according to E. Obluchinskaya (2008) and mannitol was used as reference [43].

Alginate content was measured at 400 nm and 450 nm by reaction of 3,5-dimethylphenol with sulfuric acid [116]. Alginate was used as a standard.

Laminaran content in the samples was estimated by measuring the concentration of reducing sugars (as glucose and glucose oligomers) after hydrolysis. Reducing sugars were determined spectrophotometrically at 500 nm by the glucose oxidase method [117]. Glucose was used as a standard.

Results are expressed as mg/g per DW and all measurements were performed in triplicate to ensure accuracy.

### 3.4. Analysis of Total Phenolic, Total Flavonoids, and Antiradical Activity

To analyze total phenolic content (TPhC), total flavonoid content (TFC), and DPPH scavenging activity, *A. nodosum* samples were extracted in triplicate [118] with minor modifications. Briefly, the seaweed samples (2 g) were extracted three times with 50 mL aqueous MeOH (60% *v*/*v*) in a dark place at room temperature for 24 h under continuous stirring at 200 rpm on rotator Multi Bio RS-24 (Biosan, Riga, Latvia). Afterward, the mixtures were centrifuged at 3500 rpm for 10 min, filtered (Whatman filter paper N 1), and combined. The filtrate was concentrated to dryness under reduced pressure using a rotary evaporator IR-1m (PJSC Khimlaborpribor, Klin, Russia) to remove MeOH, and the residue dissolved in 25 mL volumetric flasks with 60% *v*/*v* aqueous MeOH and filtered before use for analysis of TPhC, TFC, and DPPH scavenging activity.

TPhC in *A. nodosum* extracts was analyzed spectrophotometrically (Shimadzu UV 1800 spectrophotometer, Shimadzu, Kyoto, Japan) at 750 nm using Folin–Ciocalteu reagent [119]. TPhC was determined as mg phloroglucinol equivalent (PhE) per g DW.

TFC was measured using spectrophotometry [118,120]. The absorbance of the test solutions was recorded at 415 nm using a Shimadzu UV 1800 (Shimadzu, Kyoto, Japan) UV-Vis spectrophotometer TFC was expressed as quercetin equivalents (QE) per g DW.

DPPH scavenging activity was analyzed according to W. Brand-Williams et al. [121] with minor modifications [36]. The absorbance of the obtained solutions was measured at 517 nm.

The percent DPPH scavenged by each different samples was calculated according to Equation (1):(1)$$DPPH\;scavenging\;activity\;(\%)=\frac{A_{control}-A_{sample}}{A_{control}}\times 100$$
where A*_control_* stands for the absorbance of the control, and A*_sample_* is the absorbance of the sample solution reaction at 30 min.

The percentage of remaining DPPH-radicals was plotted against the sample/standard concentration to obtain IC_50_ value, which represents the concentration of the extract or reference antioxidant (mg/mL) required to scavenge 50% of the DPPH-radical in the reaction mixture [66].

### 3.5. Element Analysis

The samples of *A. nodosum* were extracted by the method [119]. The elements were analyzed on PerkinElmer^®^ Optima™ 8000 inductively coupled plasma optic emission spectrophotometer (ICP-OES) (PerkinElmer, Inc., Shelton, CT, USA). Details are described previously [36]. Instrumental parameters were as described by É. Flores et al. [122]. The concentration of the elements (mg/kg) was calculated according to Equation (2):(2)$$X=\frac{C_{calib}\times V\times 1000}{m}$$
where *C_calib_*—element concentration from the calibration, mg/L; *V*—volumetric flask, L; *m*—sample weight, g.

For the evaluation of the accuracy of the method, a reference sample of Cu was added to *A. nodosum* sample. The mean recovery value of Cu was 94–104%.

### 3.6. Metal Pollution Index

The metal pollution index (MPI) [123] represents the contribution of all the elements detected and calculated according to Equation (3):(3)$$MPI={\left(M_{1}\times M_{2}\times \ldots \times M_{n}\right)}^{1/n}$$
where *Mn* is the concentration of the metal *n* in the sample in mg/kg.

### 3.7. Assessments of Human Health Risk

The risk to human health of elements in *A. nodosum* samples was assessed using the target hazard quotient (THQ) and hazard index (HI) recommended by USEPA (2020). The indices were calculated according to Equations (4)–(6) below [124,125].
(4)$$Estimated\;Daily\;Intake\;\left(EDI\right)=\frac{Ci\times CR\times EF\times ED}{BW\times AT}$$
(5)$$Target\;Hazard\;Quotient\;\left(THQ\right)=\frac{EDI}{RfD}$$
(6)$$Hazard\;Index\;\left(HI\right)=\sum_{k=1}^{n=k}THQ$$
where *Ci* is the mean concentration of each element in the sample (mg/kg); *CR* is the consumption rate (0.0052 kg); *EF* is the exposure frequency (250 days); *ED* is the average exposure duration (70 years); *BW* is the average body weight (70 kg and 31.9 kg for adult and child); and *AT* is the average lifetime (72.59 years) [100]. There is no fixed consumption rate of seaweeds in Russia. As a result, the consumption rate has been considered in different studies [125]. *RfD* is the recommended oral reference dose [126].

The lifetime cancer risk (LTCR) of a population exposed to potential carcinogens in ingested seaweed was estimated according to Equation (7) below [127]:(7)$$Lifetime\;Cancer\;Risk\;\left(LTCR\right)=EDI\times CSF\times AT$$
where *CSF* is the cancer slope factor and represents the upper estimate of the slope of the dose-response curve in the low-dose range [111].

### 3.8. Statistical Analysis

Statistical analysis was performed with STATGRAPHICS Centurion XV (StatPoint Technologies Inc., Warrenton, VA, USA). Data in figures and error bars are expressed as mean ± standard deviation (SD). Differences between means were analyzed by ANOVA test and Tukey’s post hoc test. Differences were considered significant at *p* < 0.05. Pearson correlation coefficient was used to determine the relationship between representative compound content and antioxidant capacity. Multivariate data analysis was performed using multiple regression and partial least squares coefficient methods.

## 4. Conclusions

In this study, we investigated the biochemical variability, anti-radical effects, and estimated health risks of Arctic *A. nodosum* (Linnaeus) Le Jolis harvested from the Irminger Sea, Norwegian Sea, and Barents Sea during different reproductive phases. The content of major carbohydrates in *A. nodosum* (fucoidan, mannitol, alginate, and laminaran) ranged from 347 to 528 mg/g DW, depending on collection station. The proportion of the two main structural monosaccharides of fucoidan (fucose and xylose) differed significantly between seawater and reproductive stages, reaching a maximum in the BS sample during the fertility phase. The content if total polyphenols and flavonoids in *A. nodosum* samples and was increased by following order: BS < IS < NS. The antiradical activity of seaweeds was correlated with polyphenol content rather than polysaccharide content. Several toxic elements (Pb, Cd, Cr, or Ni) were not detected in most samples of *A. nodosum* from various seas of the Arctic region or were below the LOQ. The calculated low THQ and HI values evidence that regular consumption of Arctic *A. nodosum* will not induce carcinogenic risk to adults and children. We believe that our data will support the interest in *A. nodosum* as a valuable source of nutrients for the food and pharmaceutical industries utilization.

## Figures and Tables

**Figure 1 marinedrugs-22-00048-f001:**
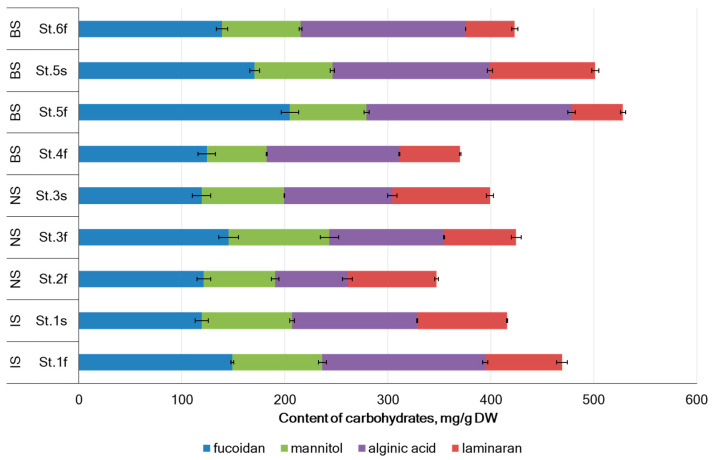
Main carbohydrates content in *A. nodosum* collected in Barents Sea (BS), Norwegian Sea (NS), and Irminger Sea (IS). Means ± SD, *n* = 3.

**Figure 2 marinedrugs-22-00048-f002:**
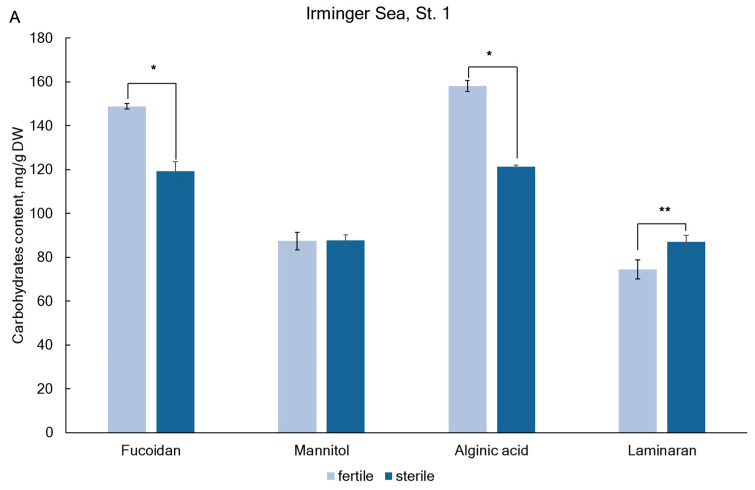
Content of carbohydrates in *A. nodosum* in sterile and fertile reproductive phases collected in Irminger Sea (**A**), Norwegian Sea (**B**), and Barents Sea (**C**); * *p* < 0.01 compared fertile phase with sterile phase. ** *p* < 0.05 compared fertile phase with sterile phase.

**Figure 3 marinedrugs-22-00048-f003:**
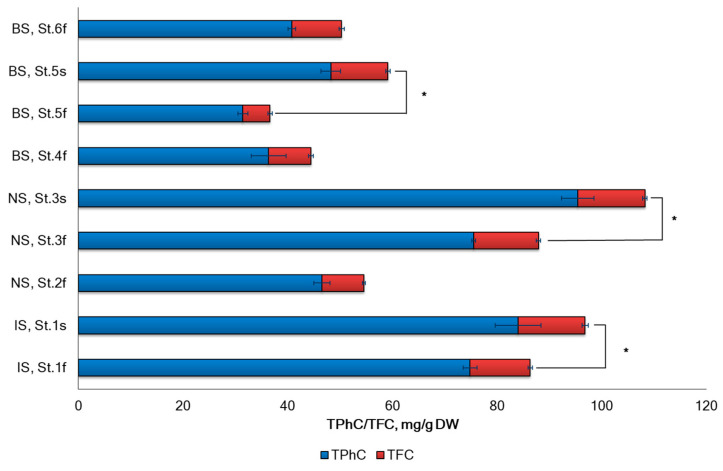
The levels of TPhC and TFC from *A. nodosum* from seas of the Arctic region. Irminger Sea (IS), Norwegian Sea (NS), and Barents Sea (BS) (errors bars for SD at *n* = 3). St. 1–St. 6—the sampling stations (details in Section 3.1). * *p* < 0.05 compared fertility with sterility.

**Figure 4 marinedrugs-22-00048-f004:**
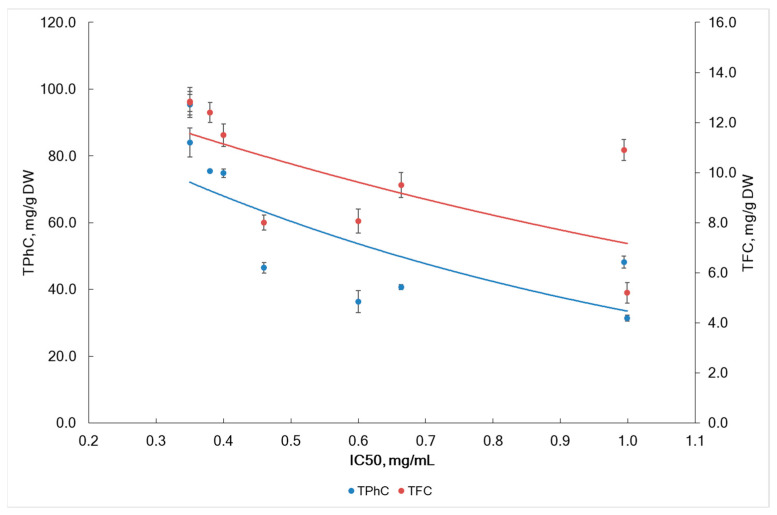
The correlation between IC_50_ and TPhC and TFC in *A. nodosum* collected in different geographical locations.

**Figure 5 marinedrugs-22-00048-f005:**
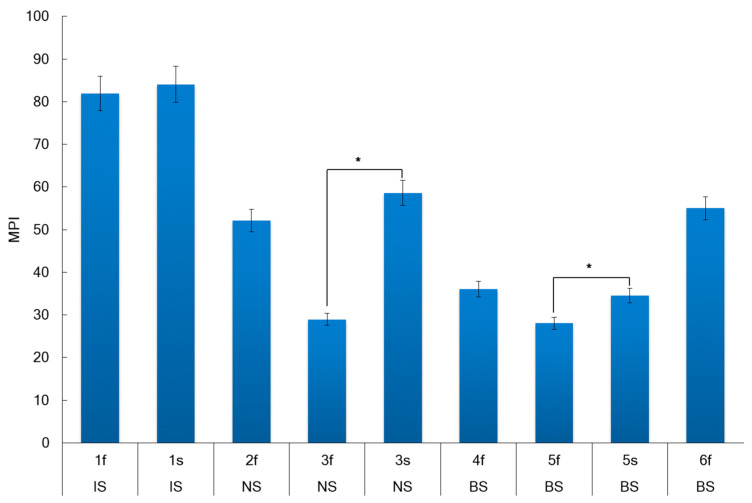
The metal pollution index (MPI) for *A. nodosum* samples collected at different regions of the Arctic. Irminger Sea (IS), Norwegian Sea (NS), and Barents Sea (BS) (errors bars for SD at *n* = 3). St. 1–St. 6—the sampling stations (details in Section 3.1). * *p* < 0.05 compared fertility with sterility.

**Figure 6 marinedrugs-22-00048-f006:**
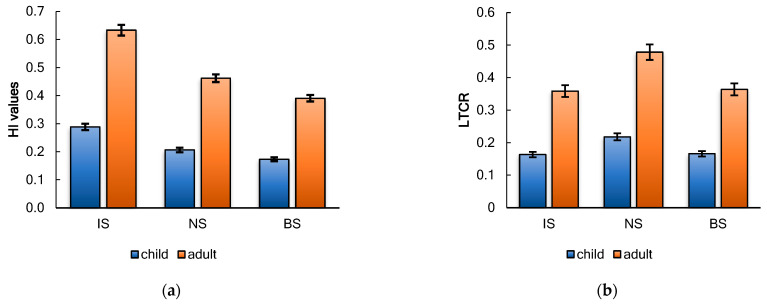
Noncarcinogenic (**a**) and carcinogenic (**b**) risks for *A. nodosum* collected in the seas of the Arctic region.

**Table 1 marinedrugs-22-00048-t001:** Fucose and xylose contents in *A. nodosum* (mean ± SD, *n* = 3).

Sea, Station	Reproductive Phase	Fucose, mg/g DW	Xylose, mg/g DW	Fucose:Xylose Proportion
IS, St. 1	Fertile	74.4 ± 1.2 *	12.4 ± 0.3 *	1.0:0.2 *
	Sterile	59.6 ± 2.9	15.9 ± 1.2	1.0:0.3
NS, St. 2	Fertile	60.6 ± 0.7	15.8 ± 0.3	1.0:0.3
NS, St. 3	Fertile	72.6 ± 2.9 *	12.0 ± 0.9 *	1.0:0.1 *
	Sterile	59.5± 1.8	15.8 ± 1.1	1.0:0.3
BS, St. 4	Fertile	62.1 ± 1.3	16.7 ± 1.0	1.0:0.3
BS, St. 5	Fertile	102.4 ± 0.8 *	22.3 ± 0.4	1.0:0.2 *
	Sterile	85.3 ± 2.5	24.5 ± 2.0	1.0:0.3
BS, St. 6	Fertile	69.4 ± 0.4	16.6 ± 0.4	1.0:0.2

Irminger Sea (IS), Norwegian Sea (NS), and Barents Sea (BS). * *p* < 0.05 compared fertility with sterility.

**Table 2 marinedrugs-22-00048-t002:** The amounts of tested elements (mg/kg DW) in *A. nodosum* collected in Arctic (mean ± SD, *n* = 3).

Element	LOQ	Mean ± SD	Range (Min–Max)	IS, St. 1		NS, St. 2	NS, St. 3		BS, St. 4	BS, St. 5		BS, St. 6
				Fertile	∆	Fertile	Fertile	∆	Fertile	Fertile	∆	Fertile
				Sterile			Sterile			Sterile		
Al	1.6	60.2 ± 55.6	16.3–168.6	137.7 ± 21.7	↑	33.6 ± 5.4	16.3 ± 10.5	*	36.8 ± 1.5	22.0 ± 1.2	–	36.1 ± 0.1
				168.6 ± 43.5			70.5 ± 11.6	↑		20.0 ± 1.2		
As	6.3	32.2 ± 7.9	21.5–46.6	26.7 ± 2.0	↑	46.6 ± 1.8	28.1 ± 2.4	*	36.0 ± 1.3	21.5 ± 0.8	*	34.0 ± 0.7
				30.8 ± 2.5			40.5 ± 4.8	↑		25.2 ± 1.2	↑	
Ba	0.016	6.6 ± 1.3	5.2–8.6	5.23 ± 0.11	*	6.4 ± 0.3	5.3 ± 1.6	–	6.3 ± 0.2	5.6 ± 0.3	*	7.4 ± 0.1
				8.55 ± 0.06	↑		6.1 ± 0.2			8.6 ± 0.1	↑	
Ca	1.9	15,595 ± 5760	10,973–26,916	17,591 ± 544	*	12,256 ± 195	10,973 ± 106	*	12,377 ± 29	12,445 ± 56	*	11,453 ± 233
				26,916 ± 813	↑		23,173 ± 199	↑		13,202 ± 27	↑	
Cd	0.23	<LOQ	<LOQ	<LOQ		<LOQ	< LOQ		< LOQ	< LOQ		<LOQ
Co	0.12	0.70 ± 0.38	0.31–1.50	1.50 ± 0.04	*	0.64 ± 0.08	0.60 ± 0.05	↑	0.93 ± 0.09	0.31 ± 0.01	–	0.35 ± 0.05
				0.84 ± 0.02	↑		0.77 ± 0.12			0.33 ± 0.04		
Cr	0.13	<LOQ	<LOQ	<LOQ		<LOQ	<LOQ		<LOQ	<LOQ		<LOQ
Cu	0.37	3.17 ± 2.61	<LOQ–7.31	7.31 ± 0.90	*	0.67 ± 0.03	<LOQ	*	<LOQ	<LOQ	<LOQ	<LOQ
				2.33 ± 0.02	↑		1.59 ± 0.82	↑		<LOQ		
Fe	0.098	90.0 ± 47.7	23.5–179.7	179.7 ± 25.2		89.1 ± 4.2	23.5 ± 10.1	*	83.9 ± 1.7	53.7 ± 1.7	*	88.9 ± 0.5
				149.1 ± 48.6	↓		77.7 ± 5.0	↑		64.1 ± 3.2	↑	
Mg	1.7	9969 ± 933	8804–11,709	11,709 ± 101	*	9867 ± 37	9159 ± 57	*	9252 ± 27	8804 ± 199	*	10,598 ± 23
				10,091 ± 14	↓		9434 ± 80	↑		10,810 ± 615	↑	
Mn	0.058	14.0 ± 8.8	7.8–33.4	33.4 ± 0.1	*	12.6 ± 0.2	8.2 ± 3.2	–	8.4 ± 0.1	7.8 ± 0.1	*	10.7 ± 0.5
				24.1 ± 0.6	↓		10.9 ± 1.2			9.9 ± 0.9	↑	
Ni	0.3	<LOQ	<LOQ	<LOQ		<LOQ	<LOQ		<LOQ	<LOQ		<LOQ
Pb	4.6	<LOQ	<LOQ	<LOQ		<LOQ	<LOQ		<LOQ	<LOQ		<LOQ
Rb	0.55	19.2 ± 6.2	10.0–28.6	18.2 ± 1.5	↑	28.6 ± 0.9	16.9 ± 1.6	*	10.0 ± 0.6	11.5 ± 1.8	*	25.7 ± 0.4
				20.8 ± 0.5			19.7 ± 1.8	↑		24.0 ± 2.9	↑	
Sr	0.026	723 ± 131	564–984	743 ± 10	*	759 ± 14	570 ± 31	*	616 ± 6	565 ± 20	*	736 ± 9
				984 ± 19	↑		729 ± 39	↑		805 ± 12	↑	
Zn	0.17	35.2 ± 14.1	25.8–71.7	25.8 ± 0.7	*	32.0 ± 2.0	34.4 ± 5.1	–	33.2 ± 0.6	26.3 ± 1.0	↑	28.4 ± 1.1
				71.7 ± 0.9	↑		34.0 ± 2.0			30.9 ± 2.6		
Ca/Mg ratio	-	1.56	-	1.5	*	1.24	1.2	*	1.34	1.41	↓	1.08
				2.67	↑		2.46	↑		1.22		

LOQ—limit of quantification; ∆—the concentration change; (↑ increase in concentration at sterility reproduction phase comparing with fertility phase; ↓ decrease in concentration at sterility reproduction phase comparing with fertility phase; comparing with sterility phase); “–“ without change; St. 1–St. 6—the sampling stations (details in Section 3.1). * *p <* 0.05 compared fertility with sterility.

**Table 3 marinedrugs-22-00048-t003:** Element concentrations, their daily dose for *A. nodosum* from different seas of the Arctic region, and comparison with daily dose risk estimators for a 70 kg man and with Nutritional Requirements.

Element	Sampling Site with a Maximum Concentration	Mean–Max Concentration (mg/kg)	Single Dose for 3.3 g Consumption (mg/Day)	Daily Dose for 12.5 g Consumption (mg/Day)	Daily Dose from Risk Estimators	Daily Nutritional Requirements
Al	IS, St. 1	60.2–168.6	0.20–0.56	0.75–2.11	70 ^1^	10 ^6^
As (total)	NS, St. 2	32.2–46.6	0.11–0.15	0.40–0.58	0.15 ^1^ (inorganic) 40 ^2^ (total)	5.0 ^7^
Ba	IS, St. 1	6.6–8.6	0.02–0.03	0.08–0.11	200	0.75 ^6^
Ca	IS, St. 1	15,595–26,916	51–89	195–337	2500 ^3^	1000 ^4^
Co	IS, St. 1	0.7–1.5	0.0023–0.0049	0.0087–0.0187	30 ^6^	10 ^6^
Cu	IS, St. 1	2.4–7.3	0.0079–0.0241	0.0299–0.0914	5 ^3,6^	0.9 ^5^/1.0 ^6^
Fe	IS, St. 1	90–180	0.30–0.59	1.12–2.25	45 ^6^	10 ^4,6^
Mg	IS, St. 1	9969–11,709	33–39	125–146	800 ^6^	400 ^6^
Mn	IS, St. 1	14.0–33.4	0.046–0.110	0.18–0.42	11 ^6^	2.7 ^4^/2.0 ^6^
Rb	NS, St. 2	19.2–28.6	0.063–0.094	0.24–0.36	200	2.2 ^6^
Sr	IS, St. 1	723–924	2.39–3.35	9.0–12.3	11 ^6^	1.9 ^6^
Zn	IS, St. 1	35.1–71.7	0.12–0.24	0.44–0.90	25 ^3^/40 ^6^	12 ^4, 6^

^1^ PTWI—provisional tolerable weekly intake; ^2^ [Directive 2002/32/EC]; ^3^ UL—tolerable upper intake level; ^4^ PRI—population reference intake; ^5^ AI—adequate intake; ^6^ [94]; ^7^ [95].

**Table 4 marinedrugs-22-00048-t004:** Characterization of collection sites of *A. nodosum*.

Sea Area	Sampling Site	Coordinates	Station	Reproductive Phases	Mean Water Temperature, °C	Range of Salinity, ‰
Irminger Sea	Fossvogur Bay	64.120887 N 21.930663 W	St. 1	Fertile	13.9	29.5–30.1
				Sterile	4.0	34.9–35.5
Norwegian Sea	Ringvassøya Island	69.815097 N 19.027894 E	St. 2	Fertile	10.6	33.7–34.3
Norwegian Sea	Cape Sydspissen	69.627168 N 18.912621 E	St. 3	Fertile	10.6	33.7–34.3
				Sterile	6.3	34.8–35.2
Barents Sea	Teriberskaya Bay (Korabelnaya Bay)	69.173088 N 35.168468 E	St. 4	Fertile	11.2	14.7–15.5
Barents Sea	Teriberskaya Bay (Zavalishina Bay)	69.184068 N 35.259487 E	St. 5	Fertile	9.1	19.9–20.7
				Sterile	4.2	24.8–26.0
Barents Sea	Dalnezelenetskaya Bay	69.117150 N 36.070790 E	St. 6	Fertile	10.3	31.0–32.0

## Data Availability

All data generated or analyzed during this study are included in this published article.

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
