# Peer review of "Ascophyllum nodosum (Linnaeus) Le Jolis from Arctic: Its Biochemical Composition, Antiradical Potential, and Human Health Risk"

_marinedrugs, 2024, doi:10.3390/md22010048_

Round 1

Reviewer 1 Report

Comments and Suggestions for Authors

The authors present an interesting study about the dietary value of artic Ascophyllum nodosum from three locations. The study is interesting in its current state because it seeks to find new sources of nutrients from marine resources.

There are, however, issues with the article.

1. The entire manuscript should be carefully read and corrected for typos, punctuation, capital letters in the middle of sentences, and lack of italics on species names.

2. Line 295 – 297: The sentence is mostly correct in structure and reporting statistical findings, but there seems to be a contradiction. The authors mention a "strong correlation," which usually implies a direct (positive) relationship, but then they state that this correlation is negative, suggesting an inverse relationship. Please clarify.

3. Line 347 - 355: The phrase "tens (calcium), hundreds (bromine, chromium) and thousands (iodine, zinc, barium) times higher" is understandable but could be clearer. Consider rephrasing to "the concentration of some elements, like calcium, can be tens of times higher; for bromine and chromium, it's hundreds of times; and for elements like iodine, zinc, and barium, it's even thousands of times higher than in seawater." "Most of ten" should be "most often." The text mentions "polysaccharides" and "ion exchange processes" but does not explain why these might be relevant to metal accumulation. A brief explanation would enhance understanding.

4. Lines 410 – 413: the sentence is repeated.

5. Regarding the mineral content, there are also several health-related ratios, such as the Na/K, Ca/Mg, or Ca/P ratios. 

Comments on the Quality of English Language

The authors demonstrate a reasonable understanding of English; however, the manuscript needs a comprehensive review for typographical and grammatical errors.

Author Response

Dear  reviewers!

Thank you very much for close evaluation of our manuscript and your comments and suggestions, which allowed us to improve the quality of paper. Our comments and responses are below. All changes in text of the manuscript are marked in red color.

We hope that in the revised form the manuscript will be accepted for the publication in Marine Drugs.

Reviewer #1

The authors present an interesting study about the dietary value of artic Ascophyllum nodosum from three locations. The study is interesting in its current state because it seeks to find new sources of nutrients from marine resources.

There are, however, issues with the article.

Q1. The entire manuscript should be carefully read and corrected for typos, punctuation, capital letters in the middle of sentences, and lack of italics on species names.

A1. Thanks for your recommendations. We rewrite all manuscript and corrected typos, punctuation, and italics in the names of algae species.

Q2. Line 295 – 297: The sentence is mostly correct in structure and reporting statistical findings, but there seems to be a contradiction. The authors mention a "strong correlation," which usually implies a direct (positive) relationship, but then they state that this correlation is negative, suggesting an inverse relationship. Please clarify.

A2. Thanks for your recommendations. We rephrase tis text.

Q3. Line 347 - 355: The phrase "tens (calcium), hundreds (bromine, chromium) and thousands (iodine, zinc, barium) times higher" is understandable but could be clearer. Consider rephrasing to "the concentration of some elements, like calcium, can be tens of times higher; for bromine and chromium, it's hundreds of times; and for elements like iodine, zinc, and barium, it's even thousands of times higher than in seawater." "Most of ten" should be "most often." The text mentions "polysaccharides" and "ion exchange processes" but does not explain why these might be relevant to metal accumulation. A brief explanation would enhance understanding.

A3. Thanks for your recommendations. We rephrase this text.

Q4. Lines 410 – 413: the sentence is repeated.

A4. Thanks for your recommendations. The sentence was repeated.

Q5. Regarding the mineral content, there are also several health-related ratios, such as the Na/K, Ca/Mg, or Ca/P ratios. 

A5. Thanks for your recommendations. We have added some information about the ratio of the main elements tested.

Q6. The authors demonstrate a reasonable understanding of English; however, the manuscript needs a comprehensive review for typographical and grammatical errors.

A6. Thanks for your recommendations.The manuscript has been extensively checked for typographical and grammatical errors.

Reviewer 2 Report

Comments and Suggestions for Authors

This study is meaningful and provide lots of important information and basic data. Some critical issues should be clarified as follows:

1.      In the part of Introduction, authors should provide more detail study information of compounds from Ascophyllum nodosum. Except for the fucoidan, phlorotannins, there are also some other chemicals should be mentioned.

2.      And the study meaning should be emphasized in the last paragraph in the part of Introduction.

3.      In the table 1, the number of significant digits of the data for Fucose:Xylose proportion should keep consistent with other data.

4.      In line 161, 164, the number of significant digits of % were not consistent. Please check the whole manuscript.

5.      In line 257-259, the sentence should be rewritten. And the reference from 1997 is a little old. It is suggested to change another relatively new one.

6.      In 267, A. nodosum should be italic.

7.      Fucoidan is water solubility, usually DPPH was not water solubility, therefore, it is not pretty good for using DPPH to evaluate the antioxidant activity of fucoidan. Please make sure the relative statements in the manuscript.

8.      In line356-364, I don’t think these information were useful, the comparation with previous study in 1973 was no meaning.

9.      For the part of human health risk, A. nodosum usually were used as feed to animal but not for human being, therefore, to analysis the risk for human health is necessary or not, please fully reconsider these statements in paper.

10.  The conclusion should be concise.

Author Response

Dear  reviewers!

Thank you very much for close evaluation of our manuscript and your comments and suggestions, which allowed us to improve the quality of paper. Our comments and responses are below. All changes in text of the manuscript are marked in red color.

We hope that in the revised form the manuscript will be accepted for the publication in Marine Drugs.

This study is meaningful and provide lots of important information and basic data. Some critical issues should be clarified as follows:

Q1. In the part of Introduction, authors should provide more detail study information of compounds from Ascophyllum nodosum. Except for the fucoidan, phlorotannins, there are also some other chemicals should be mentioned.

A1. Thanks for your recommendations. We have added additional information about studying the various compounds of Ascophyllum nodosum in the Sec. Introduction.

Q2.      And the study meaning should be emphasized in the last paragraph in the part of Introduction.

A2. Thanks for your recommendations. We have added additional information in the last paragraph of the Sec. Introduction.

Q3.      In the table 1, the number of significant digits of the data for Fucose:Xylose proportion should keep consistent with other data.

A3. Thanks for your recommendations. We modified the number of significant digits of data for the fucose:xylose ratio in Table 1.

Q4.      In line 161, 164, the number of significant digits of % were not consistent. Please check the whole manuscript.

A4. Thanks for your recommendations. The text was corrected.

Q5.      In line 257-259, the sentence should be rewritten. And the reference from 1997 is a little old. It is suggested to change another relatively new one.

A5. Thanks for your recommendations. We rephrase this text and added new information.

Q6.      In 267, A. nodosum should be italic.

A6. Thanks for your recommendations. The text was corrected.

Q7.      Fucoidan is water solubility, usually DPPH was not water solubility, therefore, it is not pretty good for using DPPH to evaluate the antioxidant activity of fucoidan. Please make sure the relative statements in the manuscript.

A7. We used algae extract obtained by extraction with 60% methanol. We have included more detailed extract preparation procedures for the analysis of total phenolic content (TPhC), total flavonoid content (TFC), and DPPH scavenging activity in section 3.4.

Q8.      In line356-364, I don’t think these information were useful, the comparation with previous study in 1973 was no meaning.

A8. Thanks for your recommendations. The text was corrected.

Q9.      For the part of human health risk, A. nodosum usually were used as feed to animal but not for human being, therefore, to analysis the risk for human health is necessary or not, please fully reconsider these statements in paper.

A9. Thanks for your recommendations. The part of human health risk has been supplemented and changed.

Q10.  The conclusion should be concise.

A10. Thanks for your recommendations.We have shortened Sec. Conclusion.

Round 2

Reviewer 1 Report

Comments and Suggestions for Authors

The authors have integrated and responded appropriately to all the reviewers’ comments. The article is acceptable for publication. 

Comments on the Quality of English Language

The English language is presented without any major issues.

Reviewer 2 Report

Comments and Suggestions for Authors